# Complex computation from developmental priors

Dániel L. Barabási [1] ✉, Taliesin Beynon[2], Ádám Katona[3] & Nicolas Perez-Nieves[4]

Machine learning (ML) models have long overlooked innateness: how strong pressures for survival lead to the encoding of complex behaviors in the nascent wiring of a brain. Here, we derive a neurodevelopmental encoding of artificial neural networks that considers the weight matrix of a neural network to be emergent from well-studied rules of neuronal compatibility. Rather than updating the network's weights directly, we improve task fitness by updating the neurons' wiring rules, thereby mirroring evolutionary selection on brain development. We find that our model (1) provides sufficient representational power for high accuracy on ML benchmarks while also compressing parameter count, and (2) can act as a regularizer, selecting simple circuits that provide stable and adaptive performance on metalearning tasks. In summary, by introducing neurodevelopmental considerations into ML frameworks, we not only model the emergence of innate behaviors, but also define a discovery process for structures that promote complex computations.

The diversity and specificity of animal behaviors, as well as their neural correlates, has received attention from diverse areas of study[1]. Recently, machine learning has provided key insights into the mechanics of solving complex behaviors[2,3]. However, AI frameworks do not capture the emergence of innate behaviors, as conventional models require extensive update rules and training examples to achieve desired fitness on a task[4,5]. Nevertheless, a number of complex tasks seem to be hard-coded into the development of the nervous system, such as mice responding to looming stimuli[6], hatched turtles heading out to sea, or humans recognizing face-like objects in the womb[7,8]. In cases where evolutionary pressures for survival outweigh learning, wiring embeddings evolve in order to encode crucial behaviors into the nascent connectome[4]. The brain's innate solutions have long inspired AI techniques[9], from convolutional neural networks to reinforcement learning, yet neuroevolutionary innovation has not been successfully recapitulated for the systematic discovery of powerful architectures.

In order to reproduce the selection process behind innate behaviors, we must first confront the mystery of the "genomic bottleneck": development's uncanny ability to unpack a genome in order to produce specific, task-relevant neural circuits[4,10]. The genomic bottleneck has been tangentially explored through networks with learned architectures but random weights[11], and by the artificial life community, such as through compositional pattern-producing networks (CPPNs), which model mechanisms in early development and morphogenesis to encode neuronal networks[12–16]. However, these approaches tend to focus on general mechanisms of development, rather than considering mechanisms specific to the genetic wiring of neuronal circuits[17]. Here, we utilize the Genetic Connectome Model (GCM), which provides a network formalization for how interactions between expressed proteins seed synapse formation[18,19], to incorporate neurodevelopmental priors into machine learning architectures. The Genetic Connectome Model has previously predicted the wiring rules that code for the *C. elegans* gap junction connectome[19], and similar formulations have been used to infer synaptic interaction rules in *C. elegans*[20] and *Drosophila*[21], providing mechanistic support to the GCM's quantitative formulation. Additionally, the GCM has been shown to promote structured connectivity, including feed-forward and scale-free networks, illustrating the constructive potential of the encoding[22].

In this manuscript, we explore whether neural recognition rules provide sufficient representational power to encode circuits capable of complex computations. We begin by considering neural network weights to be emergent from the well documented neuronal

[1]Biophysics Program, Harvard University, Cambridge, MA, USA. [2]Wolfram Physics Project, Cape Town, South Africa. [3]Electrical and Electronic Engineering, Imperial College London, London, UK. [4]Department of Computer Science, University of York, York, UK. ✉e-mail: danielbarabasi@gmail.com

recognition rules formalized in the Genetic Connectome Model[18,19]. Rather than refining weights directly, we improve fitness on our "behaviors" of standard machine learning tasks by updating the wiring rules that generate the Artificial Neural Networks' (ANN) weights, thereby modeling selection on evolutionary timescales[10]. In this formulation, the resulting neuronal wiring rules can generate an ANN capable of "innately" performing that behavior, without any additional direct weight updates. Thus, by mapping the genetic material passed on in evolution to individual fitness on a task, we provide a mechanistic model for the evolution of innate behaviors. When we test this model on categorization benchmarks, we find that our developmentally-inspired priors allow for representations robust enough for both meta- and transfer learning tasks, a hallmark of complex computation.

## Results

### Genetic neuroevolution formalism

The processing capabilities of neural systems arise from a mixture of neuronal partner selection, learning, and random noise. Of these, neuronal partner selection, consisting of axonal targeting and synaptic compatibility, can provide sufficient specificity for hardcoded circuits capable of supporting innate behaviors[23,24]. The cellular identity of neurons, as represented by their genetic profile, plays a crucial role in their preferred projections and synaptic partners. This mapping from

genes to connectivity is formalized in the Genetic Connectome Model (GCM)[18,19], which defines the wiring of the brain (**W**) as a function of neuronal identity (**X**) and interactions between genetic factors (**O**). The GCM aims to capture a time in a neuron's development when major components of cell differentiation, migration and targeting have already unfolded, and a neuron is in the process of deciding which cells to connect to, on which part of their neurite, and with what strength, defining a crucial level of specificity for hardwired behaviors. Specifically, the connectivity of $N$ neurons is described by the adjacency matrix **W** of size $N \times N$, with $W_{ij} = 1$ if a connection is present between neurons, and 0 otherwise. Individual neurons are identified by their expression of $G$ genes, defining a cell-identity matrix **X** of size $N \times G$, where $X_{ia} = 1$ if neuron $i$ expresses gene $a$ (Fig. 1a, blue to orange links). Interactions between expressed genes determine the wiring of the connectome, represented as the $G \times G$ matrix **O** (Fig. 1a, orange to orange links). Thus, the developmentally produced connectome can be formulated as

$$W = \mathcal{H}\left(XOX^T\right). \tag{1}$$

where $\mathcal{H}$ represents the Heaviside function that produces a binary connectivity matrix. Previous work used binary values for **W**, **X** and **O** to provide interpretable results for connectivity, genetic interactions and expression patterns[18,19]. In reality, genes (**X**) have continuous expression, interactions (**O**) may be probabilistic, and connections (**W**) vary in size and number. This prompts a relaxation of **W**, **X**, and **O** to $\mathcal{R}$, allowing for the three matrices to be continuous and differentiable, a modification important for calculating gradients in backpropagation.

Based on this extended framework, we propose a Genetic neuroEvolution Model (GEM) that utilizes the generative process of $W = XOX^T$ to move flexibly between a wiring diagram and its encoding in neural identities (Fig. 1b). We start by taking an architecture known to be effective on a task, then define the weights of the network using $W = X_i O X_o^T$ (Fig. 1b, downward blue arrows). Here, **W** corresponds to the weights of a layer, **$X_i$** stands for the genetic expression of the input neurons, and **$X_o$** represents the output neurons' gene expression. We begin with untrained **X**s and **O** chosen from a fitting random distribution. With each training batch, we backpropagate the loss through the weights (Fig. 1b, red leftward arrow), updating the **X**s and **O** using gradients calculated by PyTorch's autograd (Fig. 1b, red upward arrows). In theory, this results in a new developmental ruleset in the **X**s and **O**, which produces a more fit **W** in the next task initialization. We consider this formulation as a model of the neuroevolutionary process that underlies the emergence of innate behaviors, where generational pressures change the neurodevelopmental rules that produce the nascent wiring of an individual's neural system until sufficient task performance is reached at "birth."

We first aim to test the capacity of the GEM for encoding functional weight matrices. For this experiment, we turn to the MNIST hand-drawn digit classification task (Fig. 1c), which has well-studied accuracies for a number of architectures[25]. We aim to use the GEM to learn "developmentally initialized" network weights that have high task performance, without further direct weight updates ("learning") after initialization. Additionally, we aim to examine the parameter-accuracy tradeoff of the GEM encoding, as neurodevelopment works from a compressed representation of neuronal connectivity, read out from the genome (i.e. the genomic bottleneck)[4,10,18,19]. To minimize the number of free variables (i.e. size and number of hidden layers), we start by encoding a single-layer perceptron (SLP, a linear classifier) (Fig. 1d), consisting of $28 \times 28 + 10 = 794$ nodes (neurons) with 7840 learnable weights (synapses). A single layer neural network with weights initialized by kaiming uniform[26] performs at an accuracy of 10% without any training ("naive", equivalent to random choice), and with training it can achieve 93% accuracy ("learned", dashed line in Fig. 1c). We aim to encode the weights of this SLP using our GEM

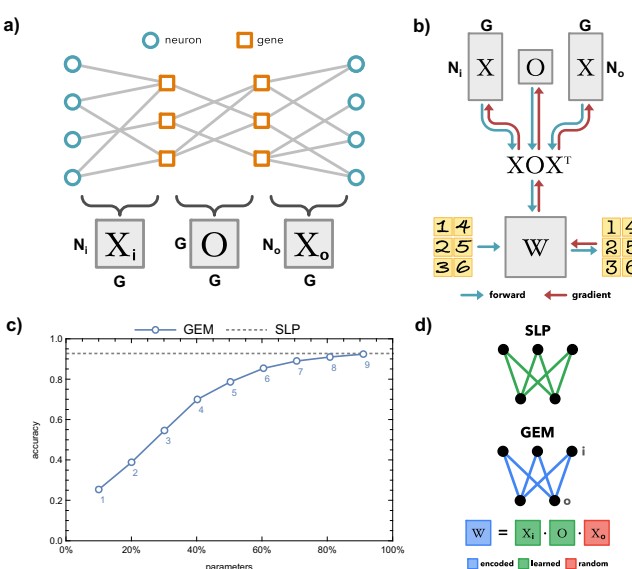

**Fig. 1 | The genetic neuroEvolution model. a** Visualization of the Genetic Connectome Model[18,19]. Matrices **$X_i$** and **$X_o$** represent the gene expression of input and output neurons, respectively. The **O** matrix corresponds to the genetic interactions that underlie neuronal partner selection. **b** Traditionally, AI techniques define an architecture (**W**), which can receive data as inputs (handwritten digits, left of **W**), and produce predictions (digital numbers, right of **W**). The weights (**W**) of the architecture can be updated (leftward red arrow) based on the distance of the predictions from known values, thereby producing a more accurate system with training. In the Genetic neuroEvolution Model (GEM), the architecture (**W**) is produced from a small set of wiring rules defined by **X**s and **O** (downward blue arrows). At each training step, the data is passed through to make predictions (rightward blue arrows). However, rather than altering the architecture weights directly, gradients are computed to update **X**s and **O** (upward red arrows). At the next training step the revised wiring rules generate a revised **W** (downward blue arrows). **c** Mean performance of the GEM on the MNIST task. The accuracy of a single-layer linear classifier (784 by 10 nodes) is shown, either with learned weights (dashed line) or weights encoded by the GEM's wiring rules (blue line, with number of genes labeled below each marker). Parameters are expressed as a percentage of a learned SLP's weights. **d** Visualization of a learned SLP and an SLP encoded by the GEM in Fig. 1c. We do not learn $X_o$, as we find it adds parameters without increasing task performance.

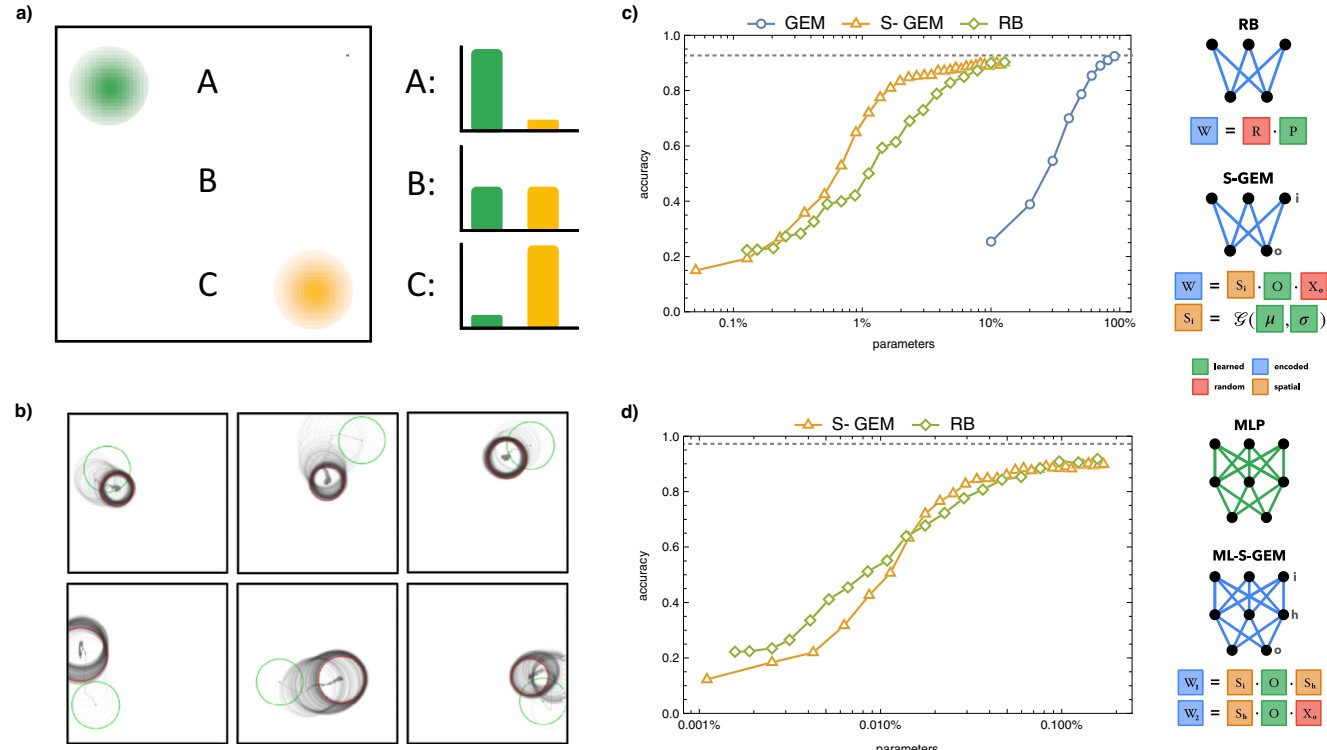

**Fig. 2 | Spatial GEM. a** In the spatial GEM, cell identities are determined by neurons' distance from 2-D gene expression gaussians. Given two spatial gene distributions, green and yellow, we have: neuron A with high expression of green and low expression of yellow, neuron B with medium expression of both genes, and neuron C with low expression of green but high expression of yellow. **b** As learning progresses, the location (μ) and dispersion (σ) of the gene expression distributions change from start (green) to finish (red). **c** Left: Parameter-accuracy tradeoff for encoding a single layer linear classifier that solves MNIST, where parameters are expressed as a % of an uncompressed SLP and reported accuracy is a mean over 10 runs. The spatial GEM (S-GEM) model significantly outperforms the GEM framework from Fig. 1c, as well as a random basis (RB) encoding. Right: Visualization of the RB and S-GEM weight encodings. **d** Left: Parameter-accuracy tradeoff for encoding an MLP that solves MNIST, where parameters are expressed as a % of an uncompressed MLP and reported accuracy is a mean over 10 runs. The multi-layer S-GEM perform comparatively to a random basis encoding. Right: Visualization of the MLP and ML-SGEM encoding.

approach. Whereas direct training has a constant $N_i \times N_o$ parameters, where $N_i$ and $N_o$ are the number of input and output neurons, respectively, a GEM-encoded SLP's parameters scale as a function of the number of genes, $G$. Specifically, the input and output layer encodings ($\mathbf{X_i}$ and $\mathbf{X_o}$) utilize $N_i$ and $N_o \times G$ parameters, while the interaction matrix $\mathbf{O}$ has $G \times G$ parameters (Fig. 1b, top). This results in a parameter scaling of $P = N_i \times G + G \times G + G \times N_o$, or $P = G(N_i + N_o + G)$, offering a compression of the weight matrix when few genes are used. In the case where $\mathbf{X_o}$ is a fixed random matrix, the parameterization of $W = X_i O X_o^T$ resembles a PCA where the first $G$ basis vectors are kept, an observation that illustrates the representational power of this biologically inspired encoding (a topic discussed further in the SI)[27].

We find that an SLP encoded by the GEM can achieve nearly 25% accuracy "at birth" when neurons "express" only 1 gene (corresponding to under 10% of the parameters of the full single-layer classifier). The accuracy increases to above 80% for $G = 5$, corresponding to 50% of the parameters of a linear classifier. The GEM achieves full 93% accuracy with 9 genes, or 90% of the baseline parameters. Thus, we find that a developmentally-motivated model of neuronal wiring provides sufficient representational power for matching the task performance of the classically trained network with fewer parameters, as well as for achieving high accuracy with a fraction of the original parameters.

**Spatial priors on neuroevolution**

Thus far, we have used a biologically-motivated model of neuronal recognition rules to derive an ML model for the emergence of innate behaviors. Next, we examine how spatial considerations for cell identity can impact neuroevolutionary search. Although constraining the

system could destabilize learning, we aim to show that complex computations can be primed by incorporating additional neurodevelopmental inspirations.

Connectivity-defining cell identity ($\mathbf{X}$ matrix) has well studied distributions in space and time[17,28–32]. To model these observations, we define a spatial GEM (S-GEM) by placing N neurons on a grid ($28 \times 28$ for an MNIST input layer), and determine a neuron's gene expression profile ($\mathbf{X}$ matrix) by its distance from $G$ 2-D gaussians with given $\sigma$ and $\mu$ (Fig. 2a, see Methods and Supplementary Methods for more details on implementation and modeling assumptions). As training progresses, we fix the locations of neurons and update the gene distributions' $\sigma$ and $\mu$, thereby learning a spatially constrained $\mathbf{X}$ (Fig. 2b). This leads to a highly compressed encoding, with 3 $G$ parameters per layer (a $\mu_x$, $\mu_y$ and single $\sigma$ for each gene), and an interaction matrix $\mathbf{O}$ with $G \times G$ parameters. Thus, an S-GEM encoded SLP has a parameter scaling of $P = G \times G + 3 G + 3 G$, or $P = G(G + 6)$, which presents a significant compression compared to $P = G(N_i + N_o + G)$ for GEM or $P = N_i \times N_o$ for standard weight updates. It is interesting to note that the parameter count does not depend on the number of neurons in the architecture, as cell identities are emergent from the local gene expression gradients.

When we use the S-GEM (Fig. 2c right) to encode the SLP from Fig. 1c, we achieve over 80% accuracy with less than 2% of the total parameters (Fig. 2c, left). The S-GEM converges towards 93% accuracy with 10% of parameters, a point at which the nonspatial GEM only performs at 25% accuracy. To better contextualize the performance of the S-GEM, we turn to the established baseline of a random basis encoding (RB, Fig. 2c left), which is a measure for the number of free

parameters needed to solve a task to a given accuracy[33]. Specifically, the RB model can be expressed as $W = RP$, where **W** is the weights of size $N_i \times N_o$, **R** is a random matrix of size $N_i \times B$, and **P** is a learned matrix of size $B \times N_o$. Here, $B$ is the number of basis vectors, and thus defines the representational power and parameter count of the RB encoding. Such an encoding can be thought of a PCA with a random basis set, hence the name of the model. We find that the S-GEM not only matches the parameter-accuracy tradeoff of the random basis model, but outperforms it at low parameter counts.

In order to confirm the robustness of the S-GEM encoding, we extend our validation to multilayer perceptrons (Fig. 2d, right). Doing so requires refining our model's assumptions. Given that gene interactions are determined by biophysical rules, and therefore are not specific to a set of neurons, we choose to learn a single **O** matrix that is shared by all encoded layers. Further, we assume that neurons have a single cell identity that determines both their input and output connectivity profiles, thus we learn a single **X** matrix for each hidden unit, using it to define its pre- and post-synaptic weights. Although neurons can express different genes in axons and dendrites, we found that learning separate input and output **X** matrices for each hidden unit only increases the number of parameters, without increasing accuracy. Thus, the total parameters of this spatial GEM scales as $P = G \times G + L \times 3G = G(G + 3L)$, as we have a single **O** matrix with $G \times G$ parameters, and each encoded layer (L) has a learned $\mu_x, \mu_y$ and $\sigma$ for each gene. As with the SLP encoding, the MLP parameter count does not scale with the number of neurons in the architecture, and offers a significant compression of the $P = \Sigma_{l=1}^{L} (N_i \times N_o)_l$ parameters it takes to represent the emergent multi-layer neural network. Although we currently restrict neurons to a grid, future work could explore an S-GEM encoding where the locations of neurons are learned, or sampled from a learned distribution, thereby naturally accommodating the addition or subtractions of neurons from a layer.

Given this definition of a multi-layer spatial GEM (ML-S-GEM, Fig. 2d right) we aim to encode an MLP of size $784 \times 784 \times 10$, where the input and hidden neurons are placed on $28 \times 28$ grids. We find that the parameter-accuracy tradeoff continues to match the benchmark of a random basis encoding, achieving above 90% accuracy with less than 0.1% of the non-encoded MLP's parameters. To provide another benchmark for comparison we turned to a less biological "genomic bottleneck" model[10], which utilizes an approach resembling hypernetworks[34] to encode the weight matrix, achieved 79% accuracy with 1038-fold parameter compression on this architecture, while the ML-S-GEM achieves above 80% accuracy with roughly 0.025% of the original MLP parameters, amounting to a 4000-fold compression. Even for this larger network, ML-GEM trains on average in 97 s, corresponding to only a minor computational burden compared to a standard MLP wall-time of 89 s.

With these successes on SLPs and MLPs, we aimed to encode the more complex LeNet-5 architecture. To do so, we derived an S-GEM parameterization of CNNs that has neurons connecting to local patches (see Methods). Our encoding approach avoids weight sharing, which is considered non-biological, thus we learn a separate kernel for each neuron in a convolutional layer, an approach that balloons LeNet-5's parameters from 61,706 to over 400,000. In this context, we aim to achieve the performance of a standard LeNet-5 while compressing the 423,038 parameters back to the realm of the original 61,706. We find that a standard LeNet-5 utilizes 61,706 parameters to achieve $98.95 \pm 0.07\%$ (mean $\pm$ std) accuracy on MNIST, while at a $2\times$ compression (29,457 parameters) our S-GEM encoded model performs at $96.7 \pm 0.2\%$ accuracy. Despite under-performing on MNIST, we find that on CIFAR-10 our S-GEM LeNet-5 ($52.5 \pm 0.6\%$ accuracy) outperforms the unencoded version ($50.6 \pm 0.5\%$ accuracy) when using an equivalent number of parameters, hinting at the potential that the S-GEM has for more complex tasks. Note that utilizing an equivalent number of parameters to the standard LeNet-5 still represents a $7\times$

compression of the non-weight-shared parameters (over 400,000 down to 61,706). In summary, we find that introducing biologically motivated constraints on cell identity not only retains the representational power of neuronal wiring rules, but also provides extreme compression of the hard-coded circuit.

Having seen GEM and S-GEM perform well on compressing working solutions to visual tasks, we aimed to understand the unique representational capacity that the approach provides. We first turned to observations from past studies which indicated that, during training, an ANN's modularity increases initially, plateaus in the middle of training, then lowers during fine-tuning[35]. We found these modularity dynamics in the GEM as well, however for the S-GEM modularity did not change significantly with training, indicating that the encodings achieve task performance by refining a set of wiring rules without significantly changing the underlying architecture (See Supplemental Methods, Fig. S6) In order to dig deeper, we examined how the models represented the characters they aimed to distinguish. Again, we found similar dynamics in the GEM and direct encodings: both within and between class distance increases at the start of training, after which within distance is reduced during finetuning (Fig. S7). In contrast, for the S-GEM between and within distance increases later, but the reduction of within distances is less drastic, if present at all, perhaps indicating that a search for a stable basis is preferred (Fig. S7). This implication is supported by a number of observations. First, we see that from the starting point (green circle) the spatial gradients make large movements early in training, but within a few training steps a good basis is found, and the Gaussian gene expression centroids only fine-tune locally (Fig. 2b). The gene gradients end up tiling the space, especially focusing on central, and informationrich, regions when G is low (as in Fig. 2b). Next, we visually examined the weights of our encoding approaches, and found that the standard MLP over-trains, learning the outlines of specific characters, a shortcoming also apparent in the random basis encoding and the standard GEM (Fig. S3–5). However, the spatial constraints of ML-S-GEM lead to a more general solution, reminiscent of the "approximate" convolutional network architectures observed by others[36], which we found promotes transfer learning (Fig. S2, Supplemental Methods). In summary, we find that the S-GEM encoding not only presents an extreme compression of network weights, but also provides an embedding that allows for the robust representation of relevant task features, even past what would be trivially expected from spatial information on a visual task.

## Developmental priming of learning

Thus far we have focused on modeling evolutionary hard-coded behaviors, where an individual has high fitness at birth. In addition to the prevalence of hard-coded traits, animals also exhibit developmentally primed behaviors, where the nascent connectome promotes the acquisition of a complex task. For instance, humans are not born with the ability to speak, nor are infants' brains predisposed to learn only their parents' native tongue. Rather, during a critical period children can rapidly acquire languages they are exposed to. This suggests that development has poised the wiring of language areas to rapidly analyze and distill sound patterns, with local wiring recontouring to encode language in the process. In this section, we show that the GEM can evolve condensed wiring rules whose rolled-out nets can flexibly, and rapidly, acquire relevant skills.

We begin by considering the meta-learning topic of few-shot classification[37,38] as a model of developmental priming of learning. Here, rather than reinforcing high fitness on a task at birth, we aim to select for embeddings that perform well within a few training steps. Developing fit embeddings in this manner can be considered as a nested optimization process: an outer "evolution" loop generates a primed **W** from **X** and **O** while an inner "learning" loop adapts the wiring to the task at hand (Fig. 3a).

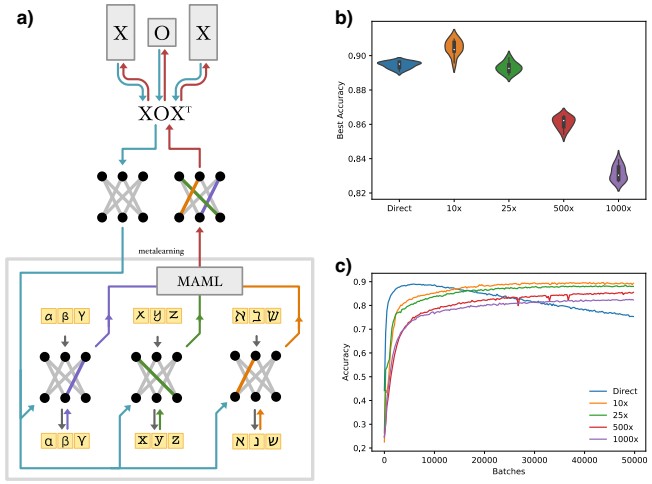

**Fig. 3 | Meta-Learning with GEM. a** Visualization of the Omniglot meta-learning pipeline. In the outer loop (top) the GEM generates an initial network, which is used for all tasks in the inner loop ("metalearning," bottom). For each classification task (Greek, Latin and Hebrew alphabets) a small number of training steps leads to diverging updates (blue, green and orange weights, respectively). The MAML algorithm allows for a meta-gradient (red arrows) to be passed back to **X** and **O** in the outer loop, thereby generating a network better suited to all tasks in the next iteration. **b** Violin plots of the best test accuracy for various models ($n = 10$ runs per model, showing median and quartile range). In order of performance, a 10x GEM achieves $90.5 \pm 0.4\%$ accuracy, a standard MLP (Direct) achieves $89.5 \pm 0.2\%$, a 25x GEM achieves $89.3 \pm 0.3\%$, a 500x S-GEM achieves $86.1 \pm 0.3\%$, and a 1000x S-GEM achieves $83.2 \pm 0.4\%$ accuracy. **c** Average accuracy of the models over 50,000 training batches ($n = 10$ runs, bands show 95% confidence interval). Although the standard MLP achieves peak performance in fewer batches, it overfits the training data, decaying in accuracy with further batches. In contrast, both the GEM (10x, 25x) and the S-GEM (500x, 1000x) compressions achieve peak performance later, but do not suffer overfitting.

Specifically, we turn to the 5-way, 1-shot Omniglot benchmark, which consists of categorizing a library of over 1000 unique characters from multiple alphabets[39]. We use the MAML algorithm[37,40] to train linear, feedforward models with two hidden layers of size 784 (see SI H: Meta-Learning on Omniglot). We find that the GEM compression can not only match the performance of a standard MLP ($89.5 \pm 0.2\%$ best accuracy over 10 runs, Fig. 3b "Direct") with a 25× compression ($89.3 \pm 0.3\%$, Fig. 3b "25×"), but can outperform it under a 10× parameter compression ($90.5 \pm 0.4\%$). We can achieve even higher compression with the S-GEM, however it comes at a slight cost to performance, with a 500× compression achieving $86.1 \pm 0.3\%$, and a 1000× compression achieving $83.2 \pm 0.4\%$ accuracy. In addition to this competitive accuracy, the GEM and S-GEM encodings do not overfit to the training data, a major drawback of the standard MLP paradigm (Fig. 3c). In summary, we find that developmentally primed encodings can provide accuracy and stability improvements over traditional ML methods for meta-learning tasks.

## Discussion

In this paper we explored a neurodevelopmental encoding for the evolution of innate behaviors. We began from a model of neuronal recognition rules, and proposed the Genetic neuroEvolution Model (GEM) in order to "evolve" a neural network that has high task performance "at birth." Next, we examined how a spatially constrained model of cell identity, reminiscent of the distributions of genes observed at the outcome of cell differentiation, can promote structured connectivity. When applied to MNIST tasks, we found that incorporating spatial priors into the GEM led to a more compact encoding, all while allowing development to prime topographic maps capable of transfer learning. Finally, we showed how utilizing the GEM

for Omniglot leads to better, and more stable performance when compared to standard MLPs of equivalent size.

Our work focused on showcasing how simple, but well documented, developmental phenomena can provide computational benefit. By further integrating neurodevelopmental principles, we expect that neuroevolutionary models can provide continued insight for machine learning, just as the visual and cognitive neurosciences have inspired the revolutionary advances of CNNs and Transformers[9,41]. For instance, the current GEM approach does not consider the complex dynamics of cell identity formation[42], which could be captured through using cellular automata to model the spatial distributions of genes in the S-GEM[43]. Another relevant direction lies in non-structured topologies: while deep neural networks utilize the simplifying assumption of layers, complex computations in the brain can arise from more intricate wiring solutions[44,45]. The computational power of non-structured neural network architectures have previously been explored[46–49], however the S-GEM can innovate on existing methods by defining a differentiable approach for neural architecture search.

Benefits may also be found by focusing in on the interplay between the genetic and activity-dependent fine-tuning of neural circuits. For instance, although the topographic mapping from retina to superior colliculus (SC) is initially established through the graded expression of receptors and ligands[50,51], the process is aided by inter-axon competition[52] and is refined by spontaneous activity before eye-opening[50,51]. Spontaneous activity is also crucial for the alignment of the topographic maps formed from retina to SC and from SC to V1[53]. Yet, spontaneous activity may play only a guiding role in circuit formation: experiments in ferrets have shown orientation selectivity emerges prior to eye-opening, a result which persist even under the silencing of spontaneous activity[54], and ex-vivo preparations, which lack spontaneous activity, still develop proper subcellular projection patterns[24]. These studies suggest that genetic mechanisms play a significant role in the maturation of computational circuits. Nevertheless, we may take inspiration from how animals grow and rewire throughout their lifetime, both in stereotyped and learning-dependent ways[55], perhaps providing insight on how ANNs can adapt continuously to changing environments and task loads. In this way, further innovations at the boundary of neurodevelopment and machine learning promises the flexible discovery of powerful inductive biases, such as CNNs and Transformers, while also demonstrating the representational power of neuroevolutionary encodings.

## Methods

### Learning on MNIST

We implemented MNIST training through the PyTorch deep learning environment. We utilized ADAM with cross entropy loss to train networks for a maximum of 30,000 batches of size 64. We tested the accuracy every 1000 batches, and included early stopping if the accuracy did not improve by more than 0.005. The accuracy was tested on a hold-out set of 10,000 digits that were not used for training. The accuracy numbers reported in the figures and text are the highest hold-out accuracy that was measured during the training run. Where multiple training runs were involved in producing a single accuracy figure, the mean was taken.

The single layer perceptron (SLP) of Fig. 1c was a single linear layer with 784 input and 10 output nodes. We trained all MLPs, encoded or not, with a single 784 hidden layer with ReLU, thereby easily comparing to concurrent models of the genomic bottleneck[10]. The only exceptions were the weights in Fig. S2a, b and Fig. S1, for which we used a hidden layer of size $8 \times 8 = 64$ to allow for easier visualizations.

When training the GEM, we begin with randomly initialized $\mathbf{X}_i$, $\mathbf{X}_0$, and $\mathbf{O}$ matrices. For each training batch, we generate the weights of a layer using $W = X_i O X_0$. We evaluate the cross-entropy loss for the batch using these weights, with which we evaluate the gradients for the **X**s and **O**. We discard the current weight matrix, update **X**s and **O**, then

generate a new weight matrix for the next batch. This is a key difference from the approach of[10], which updates the weight matrix through multiple batches, then learns by deriving rules that approximate the updated weights. In addition, while[10] alter their parameter count by changing the size of their "g-network" neural network architecture, our parameter count is determined by the number of genes expressed by each neuron. In this way, our parameter count for GEM scales as $P = N_i \times G + G \times G + G \times N_o = G(N_i + N_o + G)$, where $N_i$ and $N_o$ are the number of input and output neurons, respectively, and G is the number of genes expressed by a neuron. This means that when G = 3, each neuron is described by a vector of length 3, and an SLP can be encoded by a total of 2391 parameters, or roughly 30% of the parameters of an unencoded SLP. At this time we do not include sparsity requirements for **X**s or **O**, although this could provide relevant for further investigations.

Learning with S-GEM proceeds similarly to GEM, with weight generation and developmental rule update in each batch. However, the identities of neurons are no longer a G-length vector learned directly by backprop. Rather, we space the input and hidden neurons evenly on separate 2D grids of size 28 × 28. Then, we initialize G gene expression gaussians with parameters $\sigma$, $\mu_x$ and $\mu_y$. The gene expression of a neuron j is calculated as $X_j = e^{-(d/\sigma)2}$, where d is the distance of neuron j from $\mu_x$ and $\mu_y$. We introduce two regularizations into this process. First, we add a tanh on the $\mu$s, which ensures that we never have gene expressions that are centered outside of the grid of neurons. Second, we apply exp to the variances to ensure that the standard deviations are positive numbers. In each batch we generate the spatial **X**s, which we use to form the **W** matrix, and based on the loss update the $\sigma$s, $\mu$s and **O**, thereby learning a biologically constrained **X** (Fig. 2b). The total parameters of this spatial GEM scales as $P = G \times G + L \times 3 G = G(G + 3 L)$, as we have a single **O** matrix with $G \times G$ parameters, and each encoded layer (L) has a learned $\mu_x, \mu_y$ and $\sigma$ for each gene. In all S-GEM instances for MNIST and F-MNIST we do not use a spatial encoding for the output layer, as there is no meaningful way to place 10 nodes on a spatial grid, thus we learn an $X_0$ of size 10 directly as in GEM. In the case of Fig. 2, we simply freeze the randomly generated $X_0$ matrix, as we found learning it only adds additional parameters without increasing accuracy.

For additional implementation details, we refer the reader to the GitHub repository linked in the Data and Code Availability section.

### Developmental Convolutional Layer

Classical convolutional layers distribute neurons in a 3-dimensional grid where neurons at the same depth are said to belong to the same map and detect the same feature at different locations. This is achieved by weight-sharing across all neurons at a given map.

In our developmental convolutional layer, weights are no longer shared but constructed according to the input and output neurons' genomic expressions. Each neuron expresses a genomic identity $g(x) \in R^G$ where each component describes the expression level of a gene over a 3-dimensional parametrization of space $x \in R^3$. For each gene we have $g_i(x) \propto g_i(x) \propto 1/((x - \mu_i)^T \sigma_i)^2$ where $\mu_i, \sigma_i \in R^3$ determine the location and expression intensity of genes $i = 1, 2, ..., G$. We also have a gene interaction matrix $O \in R^{G \times G}$.

To construct the weights for each output neuron we extract the neural identity of its receptive field which is identical to that of a classical convolutional layer. That is, it consists of a $k \times k \times C_{in}$ window where $k$ is the window height and width $C_{in}$ is the depth of the input layer. Thus we are left with a matrix $X_r \in R^{G \times k2Cin}$, $r = 1, ..., R$ with $R$ being the total number of receptive fields.

Let $Y_r \in R^{Cout \times G}, r = 1, ..., R$ be the neural identities of the output neurons that share the same receptive field (i.e. output neurons at the same height and width but different depth). Then we can compute the weights for all output neurons with the same receptive field as $W_r = Y_r O X_r \in R^{Cout \times k2Cin}$. This operation can be performed efficiently for all receptive fields by using batched matrix multiplication to finally obtain the weights $W \in R^{R \times Cout \times k2Cin}$.

Note how in a classical convolutional layer the weights would be shared by all neurons irrespective of their receptive field but here, neurons on the same map but different locations may have different weights. Finally, after obtaining the weights, we can compute the output activations as in a classical convolutional layer but now using a different weight matrix for each output neuron.

### Average pooling layer

The pooling layer is a lot simpler than the Developmental Convolutional Layer. Since we need to propagate activations and neural identities to the next convolutional layer, each average pooling layer consists of two parallel pooling layers, the first one pooling the activations and the second one pooling the neural identities of the previous layer.

### Training LeNet-5

We implemented MNIST training through the PyTorch deep learning environment. We utilized ADAM with cross entropy loss to train networks for a maximum of 200 epochs broken into batches of 128 images. We tested the accuracy every epoch. Where multiple training runs were involved in producing a single accuracy figure, the mean was taken. For the 2 × compressed S-GEM LeNet-5, we utilized 150 genes per convolutional layer and 30 genes for the linear layer encodings.

Similarly, CIFAR-10 was trained over 200 epochs with 128 images in a batch. For the S-GEM LeNet-5, we utilized 229 genes per convolutional layer and 30 genes for the linear layer encodings. No additional regularizations or data augmentations were utilized in both MNIST and CIFAR-10.

### Meta-learning on omniglot

We implemented Omniglot training through the PyTorch deep learning environment. The Omniglot dataset consists of 1623 character classes total from 50 alphabets, of which 1200 classes were used for training, and the rest were reserved for testing[37]. We performed 5-way, 1-shot Omniglot with 2 inner update steps. We utilized ADAM with cross entropy loss and learning rate of $10^{-3}$ to train networks for a maximum of 50,000 batches of size 32. We learned a unique learning rate for each named parameter, allowing for more specific fine-tuning as training progressed[40]. We tested the accuracy on a hold-out set every 250 batches, averaging over 10 batches of size 64. The accuracy numbers reported in the figures and text were taken as the mean and accuracy of the highest hold-out accuracy measured in 10 individual training runs.

All trained architectures consisted of a 784 node input layer, two hidden layers of size 784, and an output layer of size 5, with ReLUs interspersed. For the GEM and S-GEM, we learned a single **O** matrix for all layers, but a unique **X** for each layer. The 10 × GEM compression corresponded to 50 genes per node, and the 25 × GEM used 20 genes per node for the **X** matrices. The 500 × and 1000 × S-GEM encodings had 43 and 28 spatial genes per layer, respectively.

### Reporting summary

Further information on research design is available in the Nature Portfolio Reporting Summary linked to this article.

## Data availability

No new datasets were generated for this manuscript. Running the central train function in the provided code automatically downloads all required datasets.

## Code availability

A GitHub repository for the Genetic neuroEvolution Model (GEM) is available at https://doi.org/10.5281/zenodo.7689897. The analyses of

the study utilized a number of publicly available software packages and datasets which could not be provided in the repository, but can be found through in-text citations.

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

## Acknowledgements
We wish to thank Gabriel Kreiman, Adam Marblestone, D´aniel Cz´egel, and the IBROSimons Computational Neuroscience Imbizo for fruitful discussions. D.L.B. was supported by NIH NIGMS T32 GM008313 and a Mind Brain and Behavior Graduate Award.

## Author contributions
All authors contributed to the design of the project and figures. D.L.B. wrote the manuscript. T.B., N.P.-N., and A.K. developed the code.

## Competing interests
The authors declare no competing interests.
