## [Peer Review File · Nature Communications]

Complex Computation from Developmental PriorsEditorial Note: This manuscript has been previously reviewed at another journal that is not operating a transparent peer review scheme. This document only contains reviewer comments and rebuttal letters for versions considered at *Nature Communications*. Mentions of prior referee reports have been redacted.

REVIEWER COMMENTS

Reviewer #1 (Remarks to the Author):

The authors have responded in a thoughtful, rigorous, and clear way to all points I raised previously. I appreciate their dedication to engaging openly in this way. *Nature Communications* is indeed a better outlet for the work than [redacted] was. I have no major remaining concerns, only to congratulate the authors on a cool piece of science, and to provide some minor tweaks suggested below.

MINOR:

- Missing period L103.
- Being a written paper, I would favor mathematical notation, (using no operator to indicate multiplication, or \times if ambiguous), over coding notation, '*', throughout (the latter typically being used for a convolution rather than multiplication in mathematical notation). I personally would also use \times for matrix sizes, but this one I suspect is just personal preference.
- When error bars are given as \pm (e.g., L194) make sure it is clear to the reader what they represent.

Reviewer #2 (Remarks to the Author):

I have reviewed this paper as a neuroscientist who can speak to the biological aspects of the paper more than to the mathematical and modeling aspects. The version of the paper available to me had been reviewed before, and the reviews, the author's responses, as well as the newly modified parts of the manuscript marked in blue were all available to me.

The previous reviewers found the links to neuroscience and developmental biology mostly unconvincing. Therefore; I will first focus on this aspect of the manuscript. The link to development is based on their previously published 'Genetic Connectome Model' – or GCM. The GCM is based on an analysis of neurons in the *C. elegans* connectome and genes expressed in these neurons. This previously led the authors to successfully predict certain features that link the genome to network features. I share the concerns of previous reviews that the GCM and s-GEM approach in the current paper do not strictly

include development. As the authors highlight, the key advance is to use a neuroevolution approach in which the GCM-based 'rules' are first updated to indirectly the weight matrix of the neural network. However, this indirect encoding does not simulate or represent a developmental process, but the endpoint of a developmental process. In response to this previous concern, the author's now argue that (in blue in the manuscript) 'The GCM aims to capture a time in development when major component of cell differentiation, migration and targeting have already unfolded and neurons are in the process of deciding which cells to connect to...' This does not make sense to me. The authors are certainly correct to point out that that such a late step in development has been proposed to involve selective 'neuro-recognition'. But, to my knowledge, all molecular interactions investigated in this respect to date turned out to have other and earlier function than just a very late step of deciding between neurons. This is really the problem: the development of 'migration and targeting' and many other developmental steps in time will result in adjacencies of neurons until synapse formation starts – and it is quite unclear whether the single time point of 'neuronal recognition' at the very end of development plays a big role. In this respect, the authors highlight the Witvliet developmental connectome paper from 2021 and papers from the Hobert lab to support the idea of development in a single step at the end. But this is in fact quite likely not correct. I would like to draw the author's attention to the most recent preprint by the Hobert lab, which was published at the time this paper was submitted, so cannot have been seen by the author's yet: <https://www.biorxiv.org/content/10.1101/2022.11.13.516316v1>

In that study, the authors provide an analysis of the *c. elegans* developmental connectomes from Witvliet 2021 and come to the conclusion that there are no neural recognition rules, and adjacencies that occur during development are sufficient to predict connectivity.

All of this just goes to say that development cannot be reduced to a single step, even if the author's try to make it the last step of 'neuronal recognition'. Correlative analysis of the genome and developmental endpoints are powerful, as their GCM shows, but they provide statistical information that correlate genes to developmental endpoints – and little more. There is no reason to assume that rules extracted from this statistical information relate specifically to a very late step of development rather than earlier steps during development, especially since this late step may not even really exist in *c. elegans*.

Taken together, I agree with the previous reviews that there is no development in the method the authors determine their priors. That's a shame, because the idea of the authors' pursuit is great and important. Yet, title, abstract and introduction do not match the work as presented. The statistical information provided by the GCM to indirectly compute priors is clearly powerful, and one can say it represents a statistical link between genes and endpoint, but these are not 'developmental priors'. The authors introduce references 18 and 28 as developmental mechanisms that motivate the development of s-GEM – but s-GEM really, really does not model any of these examples at all. Developmental gradients – and in particular alternative splicing of a molecules (*dscam*) that changes the development of dendrite branching are 100% excluded from the s-GEM approach. And they certainly do not represent only a late step of neuronal recognition after migration and targeting are done. Quite to the contrary, these are mechanisms precisely of targeting (and the change of neuronal morphologies underlying targeting- I think there is a fundamental misunderstanding of development and its reducibility here.

Yet, the s-GEM-based method of identifying priors seems to be very powerful and represents the key strength of the paper. The authors have identified priors, and a methods of updating, that allow their network model to perform benchmark tasks much more efficiently than previous approaches without

these priors. The crux of the argument lies in the origin of the priors in in what way they may resemble rules that represent biological neuronal development, as discussed above. I could not extract from the paper what 'magic' there really is in these priors, what makes them special, what makes them biological? This is where I think some improvement of the work as presented is needed and possible: the authors should provide an analysis and discussion of the priors – what makes them special and in what way are they unique? The extreme alternative (and rather scary) case would be that there are many many sets of priors that would do the same thing that can be generated without a complicated GCM. I find it impossible to judge the uniqueness or special the character of the priors used. I must assume that randomly chosen priors would perform always badly, but this critical control experiment is not included. Maybe more realistically, there is apparently something about the WJO matrix that happens to introduce properties that clearly help in a very useful way, but still have nothing to do with neuronal development. What are these properties? This might well be the golden opportunity to link the new, clearly (in terms of benchmarks) powerful approach to something that biology provides. It may not be development, but something in the statistics that link genes to networks.

Alltogether, I found this paper a difficult read that started with great promise and raised hopes, but was a bit of a let-down in the actual experiments and alleged link to neuroscience. The misconception of what constitutes development is, in my view, a major issue. On the other hand, I think there is something here and the method of determining priors identified here may reveal something important. If the authors can provide proper controls for their method to determine priors and show in what way they are special and related to biology, this could be a very interesting paper.

Reviewer #1

The authors have responded in a thoughtful, rigorous, and clear way to all points I raised previously. I appreciate their dedication to engaging openly in this way. Nature Communications is indeed a better outlet for the work than *[redacted]* was. I have no major remaining concerns, only to congratulate the authors on a cool piece of science, and to provide some minor tweaks suggested below.

MINOR:

- Missing period L103.
- Being a written paper, I would favor mathematical notation, (using no operator to indicate multiplication, or \times if ambiguous), over coding notation, '*', throughout (the latter typically being used for a convolution rather than multiplication in mathematical notation). I personally would also use \times for matrix sizes, but this one I suspect is just personal preference.
- When error bars are given as \pm (e.g., L194) make sure it is clear to the reader what they represent.

Thank you for the constructive comments throughout the review process, and for the continued attention to detail. We have implemented all the recommended changes, and defined \pm when first used (mean \pm std).

Reviewer #2

I have reviewed this paper as a neuroscientist who can speak to the biological aspects of the paper more than to the mathematical and modeling aspects. The version of the paper available to me had been reviewed before, and the reviews, the author's responses, as well as the newly modified parts of the manuscript marked in blue were all available to me. The previous reviewers found the links to neuroscience and developmental biology mostly unconvincing. Therefore; I will first focus on this aspect of the manuscript. The link to development is based on their previously published 'Genetic Connectome Model' – or GCM. The GCM is based on an analysis of neurons in the *C. elegans* connectome and genes expressed in these neurons. This previously led the authors to successfully predict certain features that link the genome to network features. I share the concerns of previous reviews that the GCM and s-GEM approach in the current paper do not strictly include development. As the authors highlight, the key advance is to use a neuroevolution approach in which the GCM-based 'rules' are first updated to indirectly [encode] the weight matrix of the neural network. However, this indirect encoding does not simulate or represent a developmental process, but [is] the endpoint of a developmental process. In response to this previous concern, the author's now argue that (in blue in the manuscript) 'The GCM aims to capture a time in development when major component of cell differentiation, migration and targeting have already unfolded and neurons are in the process of deciding which cells to connect to... ' This does not make sense to me. The authors are certainly correct to point out that that such a late step in development has been proposed to involve selective 'neuro-recognition'. But, to my knowledge, all molecular interactions investigated in this respect to date turned out to have other and earlier function than just a very late step of deciding between neurons.

We thank the Reviewer for this summary and the neuro-oriented focus.

In the quoted section we aimed to contextualize our previous work (the GCM) in the general developmental cascade. We agree that many of the proteins and genes involved in neuronal recognition have multiple functions throughout development, a key reason for why it has been difficult to study the protein-protein interactions (PPIs) underlying neuron selectivity (e.g. certain *inx* knockouts are lethal in *C. elegans*).

However, a gene having multiple functions does not rule out the known mechanistic role PPIs play in neuronal recognition and connection specificity. The clearest example is in gap junctions, where connections created by the interactions of *inx* proteins (or *cnx* proteins in vertebrates). In the case of synapses, the role of neural recognition proteins is also well studied, such as in the labs of Larry Zipursky and Josh Sanes (see joint 2020 Review in Cell), Liqun Luo (see 2021 Review in Science), and Tom Südhof (see 2018 Review in Neuron). In addition, our GCM method was developed for use in any animal, we simply applied it to *C. elegans* gap junction wiring given the high quality connectomics and gene expression data available for system, but Hobert's lab and Zipursky's lab have utilized similar approaches for synaptic PPI prediction in *C. elegans* and *Drosophila*, supporting the validity of the model.

We share the opinion that many processes in neurodevelopment not modeled by our approach do add precision, yet we additionally believe that the final stretch of neuronal recognition provides the crucial cell-type and sub-cellular specificity, a view reinforced by Josh Sanes and Larry Zipursky's 2020 Review in Cell.

We elaborate on these points throughout the remainder of our response, and have made edits to our manuscript to correct the occasional poor wording highlighted by the Reviewer, thereby focusing on the unique contribution of the GEM.

This is really the problem: the development of 'migration and targeting' and many other developmental steps in time will result in adjacencies of neurons until synapse formation starts – and it is quite unclear whether the single time point of 'neuronal recognition' at the very end of development plays a big role. In this respect, the authors highlight the Witvliet developmental connectome paper from 2021 and papers from the Hobert lab to support the idea of development in a single step at the end. But this is in fact quite likely not correct. I would like to draw the author's attention to the most recent preprint by the Hobert lab, which was published at the time this paper was submitted, so cannot have been seen by the author's

yet: <https://www.biorxiv.org/content/10.1101/2022.11.13.516316v1>

In that study, the authors provide an analysis of the *c. elegans* developmental connectomes from Witvliet 2021 and come to the conclusion that there are no neural recognition rules, and adjacencies that occur during development are sufficient to predict connectivity. All of this just goes to say that development cannot be reduced to a single step, even if the author's try to make it the last step of 'neuronal recognition'. Correlative analysis of the genome and developmental endpoints are powerful, as their GCM shows, but they provide statistical information that correlate genes to developmental endpoints – and little more. There is no reason to assume that rules extracted from this statistical information relate specifically to a very late step of development rather than earlier steps during development, especially since this late step may not even really exist in *c. elegans*.

We apologize that our writing implied that “development [occurs] in a single step at the end.”

We share the belief that migration, targeting, and other developmental steps provide crucial and specific adjacency during development, motivating the layered assumption of ANNs. We additionally believe that neural recognition provides needed final specificity for hardwired behaviors, filling in the scaffold provided by upstream processes. A paper that beautifully illustrates this is Javier Valdes-Aleman's 2021 work in *Neuron*, which showed “that neurons manage to find and connect to their partners even at abnormal locations or when silenced,” thus “neuron location, identity, and developmental activity work together to control circuit assembly.”

The Hobert lab's recent preprint provides timely insight on this topic. Yet, the preprint does not necessarily provide a nail in the coffin on the side of adjacency being sufficient for neuronal specificity, as synaptic recognition rules are mechanistically well

characterized by Larry Zipursky, Liqun Luo, and Tom Sudhof, amongst other (citations in previous section). Nevertheless, the Hobert lab's preprint allows the field to better quantify the relative contributions of adjacency and neural recognition to wiring specificity. For instance, Bill Schafer, a leading *C. elegans* biologist at Cambridge (UK), reminded us that ASEL/R are unique in that they are one of the few L/R pairs that do not form a gap junction, despite physical contact.

In this context, we have updated our references to better discuss how adjacency and neural recognition come together to provide wiring specificity. We appreciate the opportunity that the Reviewer provided to expand on this direction, as well as for sharing relevant literature by the Hobert lab.

Taken together, I agree with the previous reviews that there is no development in the method the authors determine their priors. That's a shame, because the idea of the authors' pursuit is great and important. Yet, title, abstract and introduction do not match the work as presented. The statistical information provided by the GCM to indirectly compute priors is clearly powerful, and one can say it represents a statistical link between genes and endpoint, but these are not 'developmental priors'.

We appreciate these remarks, as they have helped us to clarify the goals of the manuscript.

As related in the abstract and introduction, we turned to neurodevelopment and neuroevolution (which has led to *biological* general intelligence) in order to inspire machine learning methods development. In this manner, we used the GCM, which has been used by us, the Hobert lab, and the Zipursky lab as a model of neural recognition, in order to encode the weight matrices of ANNs.

We found that our approach led to both (1) parameter savings greater than or equal to existing priors, and (2) improvements in accuracy and stability for transfer and metalearning tasks. In this way, we aimed to showcase how developmental and ML thinking can come together, and believe that the ML task successes will encourage further interactions in this space.

Nevertheless, we believe that some confusion stems from how the word "prior" is defined across fields. In ML, this phrase has been utilized to indicate any basic bias in the training or structure of an ANN, such as convolutional neural networks or transformers. As such, "prior" could be equivalent to saying "developmentally inspired," which, in the end, our work is. We have revised the manuscript to better reflect these points, highlighting how we are using observations from neurodevelopment to motivate an ML model, rather than claiming that our ML approach in any way models all processes of neurodevelopment. In fact, in no way did we aim to capture all of development, and have now gone back and edited any section that may have hinted at this.

The authors introduce references 18 and 28 as developmental mechanisms that motivate the development of s-GEM – but s-GEM really, really does not model any of these examples at all. Developmental gradients – and in particular alternative splicing of a molecules (dscam) that changes the development of dendrite branching are 100% excluded from the s-GEM approach. And they certainly do not represent only a late step of neuronal recognition after migration and targeting are done. Quite to the contrary, these are mechanisms precisely of targeting (and the change of neuronal morphologies underlying targeting- I think there is a fundamental misunderstanding of development and its reducibility here.

We agree, and apologize for this mistake. We aimed to remove all mentions of Dscams in our previous revision (in response to Reviewer #4), as their discussion was misplaced and distracting. We simply overlooked this reference, and have replaced it now in-text, as well as any other misplaced Dscam-related references.

Yet, the s-GEM-based method of identifying priors seems to be very powerful and represents the key strength of the paper. The authors have identified priors, and a methods of updating, that allow their network model to perform benchmark tasks much more efficiently than previous approaches without these priors. The crux of the argument lies in the origin of the priors in in what way they may resemble rules that represent biological neuronal development, as discussed above. I could not extract from the paper what 'magic' there really is in these priors, what makes them special, what makes them biological? This is where I think some improvement of the work as presented is needed and possible: the authors should provide an analysis and discussion of the priors – what makes them special and in what way are they unique? The extreme alternative (and rather scary) case would be that there are many many sets of priors that would do the same thing that can be generated without a complicated GCM. I find it impossible to judge the uniqueness or special the character of the priors used. I must assume that randomly chosen priors would perform always badly, but this critical control experiment is not included. Maybe more realistically, there is apparently something about the WXO matrix that happens to introduce properties that clearly help in a very useful way, but still have nothing to do with neuronal development. What are these properties? This might well be the golden opportunity to link the new, clearly (in terms of benchmarks) powerful approach to something that biology provides. It may not be development, but something in the statistics that link genes to networks.

We thank the Reviewer for raising these points. Over the last two rounds of review we have focused our edits on (1) showing the power of GEM for transfer learning and metalearning, (2) performing benchmarking for comparison of GEM with existing priors, and (3) analyzing the ways in which GEM and S-GEM attain their representations.

The Reviewer's comments highlight that while (1) is conveyed, the unique nature of the GEM and its performance relative to benchmarks is not clearly detailed in the

current draft. We discuss these methods here, and have revised the manuscript to better highlight the comparisons to other methods.

- a) To begin, we compare our GEM and S-GEM to a random prior, as the Referee asks, in Figure 2. We utilize a Random Basis (RB) model, which has been previously been considered a measure for the number of free parameters needed to solve a task to a given accuracy. We denote the RB as $\mathbf{W} = \mathbf{R}\mathbf{P}$, where \mathbf{W} is the weights of size $N \times M$, \mathbf{R} is a random matrix of size $N \times B$, and \mathbf{P} is a learned matrix of size $B \times M$. Here, B is the number of basis vectors, and thus defines the representational power and parameter count of the RB encoding. In one sense, this can be thought of a PCA with a random basis set, hence the name of the model. We find that S-GEM either outperforms this benchmark (providing higher accuracy with equivalent parameters), or matches it (at high and low parameter counts). We now better explain the RB approach in the manuscript, and apologize for the rushed treatment this important benchmark was given previously.
- b) Of course, the results of the previous section raise the concern that S-GEM's improvement is trivial in that it's introducing a spatial prior on a visual task, where spatial information is crucial. To remedy this concern, we followed the recommendations of previous Reviewers and implemented the following checks:
 - We observed the dynamics of the spatial gene expression gradients, shown in Figure 2b. We found that from the starting point (green circle) the gradients make large movements early in training, but within a few training steps a good basis is found, and the gaussian centroids only fine-tune locally. The gene gradients end up tiling the space, with a focus on central, and information-rich, regions when G is low (such as in Figure 2b).
 - We next tested the performance of S-GEM on transfer learning, which can be done in two ways. On the one hand, we can freeze the spatial gene expression parameters and \mathbf{O} matrix, and learn only the gene expression of the output nodes (thereby updating only \mathbf{W}_2 Figure S2c, bottom right), which we consider akin to evolutionary adaptations to novel visual environments. Alternatively, can use the weights encoded by S-GEM for \mathbf{W}_1 , but update \mathbf{W}_2 directly through backpropagation, with no encoding (Figure S2c, top right), which could be thought of as on-line transfer learning within an organism's lifetime. We find that both these approaches significantly outperform the transfer learning of the standard MLP, achieving 80-90% accuracy when going from FMNIST to MNIST (Figure S2d, rows 4 and 5) and 75-80% accuracy when going from MNIST to FMNIST (Figure S2e, rows 4 and 5). As a sanity check, we wanted to ensure that this significant improvement does not come from having introduced relevant spatial information through the genetic gradients. To test this, we initialized networks

- by S-GEM but did not train them on an initial task before freezing W_1 , and found that the resulting random representation (Figure S2d,e rows 2-3) outperformed standard MLP transfer accuracy, but did not achieve equivalent performance to the transferred S-GEM approaches. In summary, we find that the transfer learning capabilities of S-GEM are augmented by pertinent information in the spatial cell identities (Figure S3), but also leverage generalized representations developed through "evolution."
- Our next investigation focused reducing the feeling of "spatial in, spatial out," where we turned to a method recommended by Reviewer 2 (Fernando et. al. GECCO 2016). The paper visually analyzes the weight matrices of an autoencoder trained on MNIST to claim that convolution is discovered by DPPNs. We considered Fernando et. al.'s approach as having established the burden of proof, and thus utilized their argument in saying that our S-GEM weights "look like visual filters," in this way generating a prototypical convolution, although we would not go as far as to say that "convolution is discovered."
 - We now better contextualize these observations throughout.
- c) Finally, the point is indeed valid that many sets of priors may exist that are equal, or better, than the GCM. Then, the question is, how do we find them? We built the GEM with the premise that neurodevelopment and evolution, given its success in finding general biological intelligence, would provide powerful priors, and our results seem to support this. We believe that incorporating additional developmental inspirations could unlock further benefits for computation, and hope that the publication of this work will inspire further studies and collaborations to explore such directions. We make these points more directly in the revised Discussion now.

Alltogether, I found this paper a difficult read that started with great promise and raised hopes, but was a bit of a let-down in the actual experiments and alleged link to neuroscience. The misconception of what constitutes development is, in my view, a major issue. On the other hand, I think there is something here and the method of determining priors identified here may reveal something important. If the authors can provide proper controls for their method to determine priors and show in what way they are special and related to biology, this could be a very interesting paper.

We appreciate this viewpoint, and the necessary light that it shone on our manuscript. While we initially wrote the study to be accessible to both neuro and ML audiences, we now realize our responses to reviewers at [redacted] and [redacted] led to a less cohesive product. We hope that the aforementioned changes have resulted in a revised manuscript that clearly conveys the motivations, model, and controls to the broad audience of *Nature Communications*.

REVIEWERS' COMMENTS

Reviewer #2 (Remarks to the Author):

The authors have thoughtfully and fully addressed my concerns. I support publication of the manuscript in its current form.

REVIEWERS' COMMENTS

Reviewer #2 (Remarks to the Author):

The authors have thoughtfully and fully addressed my concerns. I support publication of the manuscript in its current form.

We appreciate the comments of the Reviewer, and glad that they find that the revised manuscript clearly conveys the motivations, model, and controls to the broad audience of *Nature Communications*.